# Unveiling the Secrets of *Acinetobacter baumannii*: Resistance, Current Treatments, and Future Innovations

**DOI:** 10.3390/ijms25136814

**Published:** 2024-06-21

**Authors:** Andrea Marino, Egle Augello, Stefano Stracquadanio, Carlo Maria Bellanca, Federica Cosentino, Serena Spampinato, Giuseppina Cantarella, Renato Bernardini, Stefania Stefani, Bruno Cacopardo, Giuseppe Nunnari

**Affiliations:** 1Unit of Infectious Diseases, Department of Clinical and Experimental Medicine, University of Catania, ARNAS Garibaldi Hospital, 95122 Catania, Italy; federicacosentino91@gmail.com (F.C.); serenaspampinato93@gmail.com (S.S.); cacopard@unict.it (B.C.); giuseppe.nunnari1@unict.it (G.N.); 2Department of Biomedical and Biotechnological Science, Section of Pharmacology, University of Catania, 95123 Catania, Italy; uni365053@studium.unict.it (E.A.); uni318437@studium.unict.it (C.M.B.); gcantare@unict.it (G.C.); bernardi@unict.it (R.B.); 3Clinical Toxicology Unit, University Hospital of Catania, 95123 Catania, Italy; 4Department of Biomedical and Biotechnological Sciences, Section of Microbiology, University of Catania, Via Santa Sofia 97, 95123 Catania, Italy; s.stracquadanio@unict.it (S.S.); stefania.stefani@unict.it (S.S.)

**Keywords:** *Acinetobacter baumannii*, CRAB, MDR *Acinetobacter*, *Acinetobacter* treatments, *Acinetobacter* infections, cefiderocol, eravacycline

## Abstract

*Acinetobacter baumannii* represents a significant concern in nosocomial settings, particularly in critically ill patients who are forced to remain in hospital for extended periods. The challenge of managing and preventing this organism is further compounded by its increasing ability to develop resistance due to its extraordinary genomic plasticity, particularly in response to adverse environmental conditions. Its recognition as a significant public health risk has provided a significant impetus for the identification of new therapeutic approaches and infection control strategies. Indeed, currently used antimicrobial agents are gradually losing their efficacy, neutralized by newer and newer mechanisms of bacterial resistance, especially to carbapenem antibiotics. A deep understanding of the underlying molecular mechanisms is urgently needed to shed light on the properties that allow *A. baumannii* enormous resilience against standard therapies. Among the most promising alternatives under investigation are the combination sulbactam/durlobactam, cefepime/zidebactam, imipenem/funobactam, xeruborbactam, and the newest molecules such as novel polymyxins or zosurabalpin. Furthermore, the potential of phage therapy, as well as deep learning and artificial intelligence, offer a complementary approach that could be particularly useful in cases where traditional strategies fail. The fight against *A. baumannii* is not confined to the microcosm of microbiological research or hospital wards; instead, it is a broader public health dilemma that demands a coordinated, global response.

## 1. Introduction

*Acinetobacter baumannii* is considered a challenging pathogen and a significant epidemiological threat, known for both its ability to cause diseases in hospitalized patients and for developing resistance to multiple antimicrobial agents. This bacterium has increasingly become a cause for concern in healthcare settings, particularly intensive care units (ICUs), due to its capability to survive in the hospital environment and its propensity for provoking difficult-to-treat infections, including a wound to bloodstream infections and severe ventilator-associated bacterial pneumoniae (VABP), affecting patients who often already suffer from other serious medical conditions [1,2].

The increasing prevalence of carbapenem-resistant *A. baumannii* (CRAB) has exacerbated this issue, making it a critical public health and clinical priority. Carbapenems, somewhat regarded as antibiotics of last resort, are ineffective against CRAB, leading to significantly higher mortality rates—ranging from 35% to 60%—compared to infections caused by non-resistant strains [3,4]. The last ECDC report highlighted a deteriorating epidemiological landscape regarding the spread of multi-drug resistant (MDR) *A. baumannii*, with 29 EU/EEA countries reporting 10,885 *Acinetobacter* spp. isolates, more than twice the number registered in 2019. Overall, 74.5% of isolates exhibited resistance to at least one antimicrobial group, while 66.6% were resistant to three groups (fluoroquinolones, aminoglycosides, and carbapenems). Most notably, 39.9% of strains were resistant to carbapenems [5].

The economic burden is also substantial, with increased healthcare costs stemming from longer hospital stays, greater use of expensive last-resort antimicrobial agents, and heightened need for isolation and stringent infection control measures [4]. Recognizing the gravity of the situation, the World Health Organization (WHO) has classified *A. baumannii* as a critical pathogen requiring urgent development of new antibiotics [2].

Currently, only a few drugs retain some activity against CRAB, and recent guidelines recommend combination therapy to enhance their effectiveness and minimize treatment failure. However, this approach often comes with a significant pharmacological burden and suboptimal efficacy [6].

The resistance mechanisms employed by *A. baumannii* are diverse and complex, involving both intrinsic and acquired strategies that complicate treatment and clinical management. A comprehensive understanding of the resistance patterns is crucial for developing effective therapeutic and preventive measures. Key resistance pathways include the production of β-lactamases, particularly carbapenemases, which hydrolyze most β-lactams, efflux pumps which expel antibiotics from the cell, reducing their efficacy, and modifications of antibiotic target sites, which prevent drug binding and thereby negate their effects [7,8,9].

*A. baumannii*’s genetic adaptability also plays a critical role. Noteworthy, this pathogen has an exceptional ability to acquire and disseminate genetic elements, namely plasmids, transposons, and integrons, carrying resistance genes. Such genetic plasticity is fueled by its natural competence and the prevailing selective pressures in the environment, leading to the rapid spread of resistance traits within and across species [10,11,12,13].

Recent studies have also highlighted that *A. baumannii* can form a biofilm, which provides a protective niche that enhances survival and resistance in hostile conditions. Biofilms are complex communities of microorganisms highly resistant to antimicrobial agents and host immune responses, posing a formidable barrier to effective infection care [10].

In view of the above, it appears evident that *A. baumannii* is not only a model organism for studying antibiotic resistance but also a serious global health hazard that requires concerted efforts to manage and control [14]. The development of new antimicrobial agents and alternative therapies, along with strategies to inhibit resistance mechanisms, such as efflux pump inhibitors and β-lactamase blockers, are urgently needed to treat infections caused by this resilient pathogen [15]. As *A. baumannii* continues to evolve, so too must our approaches to combat this formidable enemy in the fight against antibiotic resistance.

This review aims to summarize the resistance mechanisms of *A. baumannii* and explore current and novel therapeutic options against CRAB. By understanding and addressing these mechanisms, we may ultimately be able to design more effective strategies to prevent and treat diseases caused by this cumbersome pathogen.

## 2. Resistance Mechanisms

### 2.1. Enzymatic Inactivation

#### 2.1.1. β-Lactamases

β-lactamases are enzymes produced by bacteria that confer resistance to the widely used β-lactam antibiotics such as penicillins, cephalosporins, monobactams, and carbapenems. These enzymes are one of the main mechanisms through which *A. baumannii* is able to evade the β-lactams antibacterial activity. The production of β-lactamases, especially carbapenemases, poses significant problems in clinical settings due to their ability to hydrolyze and inactivate carbapenems, which are often considered backbone antibiotics for the treatment of severe and MDR bacterial infections [7,11,16]. The genes encoding β-lactamases are often located on mobile genetic elements, i.e., plasmids, transposons, or integrons, which facilitate their horizontal transfer between bacteria. This genetic mobility is a critical factor in the rapid spread of β-lactamase-mediated resistance among *A. baumannii* populations, particularly in hospital environments where antibiotic use is high. The presence of insertion sequences such as ISAba1 adjacent to carbapenemase genes (e.g., bla_OXA-23-like genes) can promote the expression and dissemination of these resistance factors [9,13,17].


*Types of β-Lactamases Produced by A. baumannii*


**Class A**: enzymes can hydrolyze penicillins and early-generation cephalosporins. Some class A enzymes have evolved the ability to effectively hydrolyze carbapenems. An example is the KPC-type (*Klebsiella pneumoniae* carbapenemase), which, although rare in *A. baumannii*, has been identified in isolates from outbreak scenarios [18].

**Class B**: metallo-β-lactamases (MBLs), which require zinc ions for activity, are capable of hydrolyzing penicillins, cephalosporins, and carbapenems. Common MBLs in *A. baumannii* include the NDM (New Delhi metallo-β-lactamase), VIM (Verona integron-encoded metallo-β-lactamase), and IMP (imipenamase) types. MBL-producing strains are particularly challenging because they are often resistant to all β-lactams except monobactams [19].

**Class C**: AmpC β-lactamases hydrolyze cephalosporins and are usually not inhibited by clavulanic acid, a common β-lactamase inhibitor. In *A. baumannii*, AmpC is chromosomally encoded and induced in the presence of β-lactam antibiotics, promoting resistance escalation during therapy [11].

**Class D**: (OXA-type) enzymes are typically oxacillinases and have a broader substrate profile, including the ability to hydrolyze carbapenems. *A. baumannii* is particularly notorious for its production of OXA-type carbapenemases, namely OXA-23, OXA-24/40, and OXA-58, which are pivotal in mediating resistance to carbapenems, the last line of defense for many bacterial infections. Although usually harbored by plasmids, *A. baumannii* strains possess chromosomally encoded OXA-51-like β-lactamases [20]. The prevalence and diversity of OXA enzymes represent a significant epidemiological challenge in containing infections caused by *A. baumannii* [21].

#### 2.1.2. Aminoglycoside-Modifying Enzymes

Aminoglycoside-modifying enzymes (AMEs) inactivate aminoglycoside antibiotics by phosphorylation, adenylation, or acetylation, preventing them from binding to their bacterial ribosomal target and thus nullifying their bactericidal effect [9].

### 2.2. Target Modification

Target modification is a critical resistance mechanism adopted by *A. baumannii*, which involves altering the antibiotic binding site so that the drug becomes ineffective. This mechanism mainly affects β-lactams, aminoglycosides, and colistin via spontaneous or induced gene mutations and horizontal gene transfer (HGT). By modifying the targets, *A. baumannii* can thrive in the presence of antibiotics that would otherwise inhibit its growth or even kill the bacteria.


*Target Modification in β-Lactam Resistance*


As β-lactam antibiotics bind to penicillin-binding proteins (PBPs) to exert antimicrobial activity, resistance also arises from *A. baumannii* altering its PBPs, which eventually leads to the weakening of binding affinity. Modification can occur through mutations in the genes encoding PBPs or the acquisition of new PBP genes from other bacteria [7]. The altered proteins maintain their physiological function but evade inhibition by β-lactams, a phenomenon observed in several clinical isolates of *A. baumannii* [19].


*Target Modification in Aminoglycoside Resistance*


One of the most significant modifications in *A. baumannii* involves methylation of the 16S rRNA component of the 30S ribosomal subunit, hence an alteration of the binding site for aminoglycosides. As a consequence, effective antibiotic binding is hindered, and protein synthesis is disrupted. This modification is often caused by AMEs. Genes encoding these enzymes, such as *armA*, are typically located on mobile genetic elements that facilitate their spread throughout bacterial populations [21].

Enzymatic modification of antibiotics themselves, i.e., phosphorylation, adenylation, or acetylation by AMEs, also contributes to resistance. These chemical modifications inhibit the drug’s ability to bind to its ribosomal target, effectively neutralizing its bactericidal or bacteriostatic effects [7].


*Target modification in colistin resistance*


The primary mechanism by which *A. baumannii* exhibits resistance to colistin is the structural alteration of lipid A, usually by the addition of phosphoethanolamine (PEtN) and 4-amino-4-deoxy-L-arabinose (L-Ara4N). Such modifications are mediated by the *pmrCAB* operon, which is commonly activated in response to environmental signals like the presence of cationic antimicrobial peptides such as colistin. The attachment of these groups to lipid A reduces the negative charge of the lipopolysaccharide (LPS), diminishing the binding efficacy of colistin. Some strains of *A. baumannii* can develop colistin resistance through the complete loss of LPS due to mutations or altered expression of the *lpxA*, *lpxC*, and *lpxD* genes, all involved in the biosynthesis of lipid A [22]. The absence of LPS results in a deep modification of the outer membrane, which removes the primary binding target of colistin, hence conferring resistance [23].

### 2.3. Reduced Permeability and Active Efflux

Reduced permeability is a core antibiotic resistance mechanism in *A. baumannii*, predominantly impairing the activity of hydrophilic antibiotics such as β-lactams, aminoglycosides, and tigecycline. Indeed, alterations in the bacterial outer membrane impede the entry of antibiotics into the microbial cell, reducing intracellular concentrations and effectiveness [7,24].


*Alterations in Outer Membrane Proteins*


The outer membrane of *A. baumannii* features porin channels that facilitate the passive diffusion of molecules, including antibiotics, into the cell. Modifications of porins, either through expression levels or structural changes, significantly limit antibiotic uptake. For instance, loss or downregulation of the 33–36 kDa porins OmpA and CarO has been directly associated with increased resistance to carbapenems and other β-lactams [25].

Moreover, mutations in the genes encoding porin proteins can lead to altered pore sizes or charges, further decreasing membrane permeability to antibiotics. These mutations often arise from the selective pressure exerted by the clinical use of antibiotics, enabling the survival and proliferation of strains with modified porin profiles [21].


*LPS Modifications*


Changes in the composition of LPS molecules on the outer membrane also contribute to reduced permeability. Alterations can affect the overall charge and hydrophobicity of the membrane, hindering the diffusion of antibiotic molecules, particularly those that are more polar.

Specific structural changes in the lipid A component of LPS have been observed that may impact the membrane barrier function. In many cases, these changes are achieved by HGT of genes encoding enzymes capable of modifying lipid A [26].


*Capsule Production*


Many clinical isolates of *A. baumannii* produce a dense, polysaccharide-rich capsule that envelops the bacterium. This capsule acts as an additional barrier to antibiotic penetration. It physically restricts access to the outer membrane, where porin channels and other antibiotic uptake mechanisms are located. The capsule is particularly responsible for resistance to aminoglycosides and other antibiotics that require direct interaction with the outer membrane to enter the cell [27].


*Biofilm-Associated Reduced Permeability*


When *A. baumannii* forms biofilms, the cells are encased in an extracellular matrix that significantly hampers antibiotic penetration. This matrix, composed of polysaccharides, proteins, and extracellular DNA, is able to reduce the diffusion rate of antibiotics, particularly larger molecules, thus protecting the bacterial community within the biofilm [28].


*Active Efflux*


The AdeABC efflux system is best characterized in *A. baumannii*. It belongs to the resistance-nodulation-cell division (RND) family of efflux pumps and expels a wide range of antibiotics, including fluoroquinolones, tetracyclines, and chloramphenicol, lowering their intracellular concentration and efficacy. Regulatory genes *adeR* and *adeS* (encoding a two-component system) control the expression of AdeABC, and mutations in these genes can lead to overexpression and, hence, high-level resistance [12,25]. The AdeABC efflux system will be more deeply investigated in the following paragraphs. Figure 1 provides a schematic representation of all resistance mechanisms described above.

### 2.4. Biofilm-Associated Resistance

*A. baumannii* exhibits several peculiarities in biofilm formation that contribute significantly to its persistence and MDR [29,30]. Biofilms are complex, structured communities of bacteria and/or fungi encased in a self-produced extracellular polymeric matrix consisting of polysaccharides, proteins, lipids, and nucleic acids. The matrix provides a protective environment that enhances bacterial survival against the immune system and antibiotic treatments [31,32].

Biofilm formation consists of several stages, starting with initial adhesion to surfaces, followed by microcolony formation, maturation, and eventual dispersal of cells to colonize new niches. The process is regulated by numerous genes and environmental factors. Key biofilm-related genes include *bap* (biofilm-associated protein), *ompA* (outer membrane protein A), *csuE* (part of the chaperone–usher pathway), and *pgaB* (involved in polysaccharide synthesis). All play a critical role in cell adhesion, biofilm maturation, and structural stability [32,33].

Specifically, the protein Bap is essential for intercellular adhesion and biofilm growth, while OmpA facilitates attachment to epithelial cells and biofilm development on abiotic surfaces. The CsuE protein is part of a pilus assembly system crucial for initiating biofilm formation, and PgaB is involved in producing extracellular polysaccharides that constitute the matrix [28]. These components work in tandem to build robust biofilms that are highly resistant to immune cells and antimicrobial agents. Biofilms are associated with a wide range of hospital-acquired infections, including VABP, urinary tract infections, wound infections, and bacteremia. These infections are particularly challenging to treat due to the enhanced resistance of biofilm-embedded bacteria. Notably, antibiotic concentrations required to eradicate biofilms can be up to 1000 times higher than those needed to kill planktonic bacteria [28,33].

*A. baumannii* employs a sophisticated quorum sensing (QS) system, primarily the AbaI/AbaR, which is analogous to the LuxI/LuxR system found in other Gram-negative bacteria, to regulate biofilm formation. This system relies on the production and detection of acyl-homoserine lactones (AHLs), specifically N-hydroxydecanoyl-L-homoserine lactone (OHC12-HSL), as signaling molecules [34,35]. AHLs are synthesized by the enzyme AbaI and sensed by the receptor AbaR. The concentration of AHLs increases with cell density and triggers the expression of genes involved in biofilm formation and virulence when a threshold concentration is reached [36].

Autoinducer-2 (AI-2) is a universal signaling molecule used by many bacterial species, including *A. baumannii*. AI-2 is engaged in interspecies communication and boosts biofilm formation, increasing the ability of bacteria to colonize surfaces [37]. Furthermore, environmental factors, such as nutrient availability, temperature, pH, and the presence of specific ions, can also act as inducers or modulators of quorum sensing in *A. baumannii* [38,39].

Potential treatments to tackle biofilm-associated *A. baumannii* infections focus on disrupting biofilm formation, enhancing antibiotic penetration, and employing alternative antimicrobial strategies. One promising approach is the use of quorum sensing inhibitors (QSIs) and quorum quenching (QQ) enzymes, which target the bacterial communication systems underlying biofilm formation. Monounsaturated fatty acids such as palmitoleic acid and myristic acid have shown potential in reducing biofilm development by downregulating the expression of the QS system. Plant extracts and synthetic molecules have also been investigated for their ability to interfere with QS, thus attenuating biofilm growth [40,41]. Quorum quenching (QQ) enzymes, such as AHL lactonase, can degrade AHLs and disrupt QS signaling. Indeed, the application of QQ enzymes can significantly limit biofilm production as well as bacterial virulence by preventing the accumulation of AHLs. Importantly, the QQ enzyme Aii20J with AHL-lactonase activity has been shown to decrease biofilm formation and surface motility in *A. baumannii* [42] (Figure 2).

## 3. Treatment Options

### 3.1. Current Therapies

#### 3.1.1. Sulbactam-Based Regimens

Sulbactam, a first-generation narrow-spectrum β-lactamase inhibitor typically used in combination with ampicillin, is a key treatment option for *A. baumannii* infections. The effectiveness of ampicillin/sulbactam in fighting *A. baumannii* stems primarily from the sulbactam moiety, which possesses significant bactericidal activity also against β-lactamase producing strains. Interestingly, sulbactam acts through the inhibition of essential PBPs, specifically PBP1 and PBP3, which are vital for the synthesis of the cell wall in Gram-negative bacteria [43,44]. For mild to moderate carbapenem-susceptible and carbapenem-resistant infections, dosages of 3 g (2 g of ampicillin plus 1 g of sulbactam) intravenously every 6 to 4 h are recommended. In more severe cases, the dosage can be titrated to 9 g every 8 h or even up to 27 g per day as a continuous infusion for optimal exposure.

Clinical studies have shown that ampicillin/sulbactam is at least as effective as other agents like colistin and imipenem, with clinical success rates ranging from 83% to 93% [45,46]. Notably, in CRAB infections, combination therapies consisting of high doses of ampicillin/sulbactam (6–9 g per day of sulbactam component) plus at least a second antibiotic (such as cefiderocol or tigecycline) have demonstrated reduced mortality and less nephrotoxicity compared with colistin-based regimens and are suggested as a first-line regimen in recent guidelines [6]. Ampicillin/sulbactam is used even in settings with extensive drug resistance (XDR), even if the isolate is reported to be resistant, optimizing the dosage with high-dose prolonged infusion. In fact, due to its pharmacokinetic properties, high doses of sulbactam have the ability to overcome resistance mechanisms [47].

#### 3.1.2. Tigecycline

Tigecycline represents an alternative approach to value in scenarios where resistance is most prevalent. Because of its broad-spectrum activity against Gram-negative bacteria, tigecycline is a valuable addition to the antibiotic toolkit, even targeting some MDR and XDR strains of *A. baumannii* [48,49]. The recommended dosages vary depending on the severity of the clinical presentation. The regimen starts with a 100 mg loading dose followed by 50 mg every 12 h for mild carbapenem-susceptible infectious diseases, whereas in the case of more severe CRAB infections, the loading dose is increased to 200 mg, followed by 100 mg every 12 h. Clinical studies have shown mixed results, with success rates for tigecycline treatments ranging from 47% to 81% [50]. While tigecycline generally shows no significant differences in all-cause mortality compared to other antibiotics, it is associated with a lower rate of microbial eradication and tends to lengthen hospital stays. Notably, higher doses of tigecycline have been linked with improved outcomes, particularly in critically ill patients with VABP caused by drug-resistant strains. However, the emergence of tigecycline resistance during treatment has been reported at rates of around 12%, underscoring the need for careful monitoring and possibly the use of combination therapy to mitigate the development of resistance. *A. baumannii* becomes resistant primarily through the overexpression of the AdeABC efflux pump, which actively expels tigecycline from the cell, reducing its efficacy. The AdeABC efflux pump consists of three proteins: AdeA, AdeB, and AdeC. AdeB is a critical component, serving as the multi-drug transporter, while AdeA and AdeC facilitate the process by forming a structural complex that supports the function of AdeB [51]. Regulation of this efflux pump is intricately controlled by genetic elements and regulatory systems. The two-component system AdeRS is vital for the expression of AdeABC. Specific mutations in AdeRS, like A94V and S8A in AdeS or P56S in AdeR, are known to upregulate AdeABC expression, thereby increasing resistance to tigecycline. Moreover, the insertion of genetic elements such as IS Aba1 into the *adeS* gene has been proven to elevate *adeB* expression, further enhancing efflux activity and resistance [51]. Additionally, environmental pressures, such as exposure to sub-minimal inhibitory concentrations (sub-MIC) of tigecycline, can induce adaptive changes. Under these conditions, *A. baumannii* can evolve to increase the expression of the AdeABC efflux pump, thus allowing the bacteria to survive despite antibiotics [52].

#### 3.1.3. Cefiderocol

Cefiderocol, a siderophore cephalosporin, represents, up to now, a game-changing molecule in the treatment of *A. baumannii* infections, particularly in cases involving CRAB. It is highly regarded for its effectiveness against difficult-to-treat MDR organisms due to its unique mechanism of penetrating cells by mimicking iron, which is actively transported by bacteria across their cell membranes [53,54].

Recent studies highlight the efficacy of cefiderocol and its potential as a promising alternative in scenarios where traditional treatments may fail. It has shown substantial in vitro activity against CRAB and is used in severe infections where other options are ineffective or contraindicated. Although the clinical use of cefiderocol has been linked to higher mortality rates in some patient groups compared to the best available therapy, mainly due to the non-homogeneous patient population [55], other studies reported a high success rate of this molecule, especially in critically ill patients [56,57], necessitating cautious use, particularly in individuals with complex clinical profiles, as those with renal dysfunction or septic shock [58].

Importantly, resistance development during treatment has been observed, albeit at relatively low rates [59]. This underlines the need for vigilant monitoring and possibly combining cefiderocol with other agents to manage serious infections effectively and mitigate resistance risks.

The standard dose is 2 g infused over 3 h every 8 h for patients with normal renal function or mild renal impairment (creatinine clearance 60–119 mL/min). In patients with moderate renal dysfunction, the dosage is reduced to 1.5 g every 8 h and further decreased to 1 g for those with severe impairment. Ultimately, individuals with end-stage renal disease are advised to take 0.75 g every 12 h.

Clinical guidelines suggest that cefiderocol should be used particularly in severe infections where other treatments have either failed or are not tolerated. It is recommended as a component of combination therapy to potentially enhance its efficacy and reduce the odds of resistance development [54]; suitable partner drugs could be ampicillin/sulbactam, tigecycline, and intravenous fosfomycin (to which *A. baumannii* in intrinsically resistant, but it results in a synergistic activity reducing other molecules’ MIC) [45,60].

The resistance mechanism involves several genetic and phenotypic adaptations. One key factor is the presence of β-lactamases, particularly PER-like β-lactamases, and MBLs, which have been identified as significant contributors to reduced susceptibility to cefiderocol. Specifically, the PER-1 β-lactamase has been reported to increase cefiderocol MICs significantly, confirming its role in resistance [61,62].

Additionally, the downregulation of iron-uptake systems, such as the TonB-dependent siderophore receptor PiuA, has been associated with resistance occurrence [53,63]. It is worth emphasizing that mutations resulting in lower expression of the *piuA* gene greatly influence bacterial susceptibility to cefiderocol, as they compromise the implementation of the “Trojan horse” strategy, which exploits the hijacking of bacterial iron transport mechanisms to gain effective access into the cell [64].

Finally, in vitro cefiderocol testing shows several issues; thus, EUCAST does not propose a susceptibility breakpoint, making the choice of this antibiotic almost only empirical [65]. Although some authors highlighted issues related to cefiderocol testing and clinical outcomes in early studies [55], especially in critically ill patients, other authors emphasized the heterogeneous populations in those studies and the importance of adjusting for confounding factors. They noted that well-designed observational studies have shown cefiderocol to significantly reduce mortality rates compared to other treatments (best available therapies, BAT) [66]. Cefiderocol remains, at least for now, the only drug with activity against CRAB [67]. Therefore, further studies should be conducted to evaluate its full potential, especially in monotherapy.

#### 3.1.4. Eravacycline

Eravacycline is a new synthetic fluorocycline recently approved by the FDA and the EMA for the treatment of complicated intra-abdominal infections (cIAI) in adult patients [68]. It is structurally similar to tigecycline but has two unique chemical modifications that confer in vitro activity against most Gram-positive and Gram-negative organisms, including CRAB, except for *P. aeruginosa* [68,69]. It can be administered intravenously and orally and exerts its antibacterial activity by reversibly binding to the bacterial ribosomal 30S subunit [70]. A double-blind RCT (IGNITE 1) compared eravacycline used at a dose of 1 mg/kg of body weight intravenously every 12 h with ertapenem 1 g every 24 h for the treatment of cIAIs requiring surgery or percutaneous drainage [71]. The primary endpoint was the clinical response rate at the test of cure in the microbiologically modified intent-to-treat (mITT) population. In the mITT population, the clinical cure was achieved in 87% (235/270) and 89% (238/268) of patients receiving eravacycline and ertapenem, respectively (difference −1.80%, with 95% CI from −7.4% to 3.8%). The study confirmed the achievement of non-inferiority in all three primary study populations. The double-blind IGNITE 4 RCT compared intravenous eravacycline with meropenem for the treatment of cIAI requiring surgical or percutaneous intervention [72]. The primary endpoint was clinical cure at the test of cure. In the primary study population (micro-ITT), non-inferiority was achieved with clinical cure rates of 91% (177/195) and 91% (187/205) in the eravacycline and meropenem arms, respectively (difference −0.5%, with 95% CI from −6.3% to 5.3%). In pooled analyses of IGNITE1 and IGNITE4, eravacycline confirmed its high microbiological response rate also in patients with *A. baumannii* cIAIs [73]. Overall, eravacycline can be considered an important option for the treatment of cIAI due to the difficulty of treating *A. baumannii* (DTT-AB). Its good safety profile, the availability of an oral formulation, and its potential use in VAP as an alternative to β-lactams (e.g., in allergic patients) are very attractive for DTT-AB infections. Kunz Coyne et al. [74] reported a retrospective, observational study involving 416 patients who received eravacycline for at least 72 h across 19 medical centers in the U.S. The primary outcome was clinical success, defined as survival without microbiological recurrence 30 days post-therapy and clinical improvement within 96 h of initiation. Eravacycline was predominantly used to treat infections caused by *Enterobacterales* spp. (42.3%), *Enterococci* spp. (24%), and *Acinetobacter* spp. (23.3%). Notably, 47.4% of *Acinetobacter* infections treated were CRAB. Clinical success was observed in 75.7% of patients. The Clinical and Laboratory Standards Institute (CLSI) and the European Committee on Antimicrobial Susceptibility Testing (EUCAST) do not have breakpoint data available for eravacycline in CRAB infections due to insufficient evidence [75,76]. Real-world stronger data are needed to better understand the place in therapy of this promising molecule.

### 3.2. Innovative Approaches

#### 3.2.1. Novel Beta-Lactams/Beta-Lactamase Inhibitor Combinations (BLICs)

##### Sulbactam/Durlobactam

Sulbactam/durlobactam (SD) is an innovative treatment regimen specifically designed to address infections caused by CRAB. The widespread dissemination of β-lactamases among *A. baumannii* strains has compromised sulbactam combination therapy effectiveness, thus warranting new antimicrobial strategies.

As a diazabicyclooctane β-lactamase inhibitor, durlobactam protects sulbactam from degradation by a broad range of serine β-lactamases, comprising those commonly found in *A. baumannii* (e.g., OXA-type carbapenemases), thereby restoring and amplifying its antibacterial activity. Clinical trials have reported significant improvements in the susceptibility of CRAB isolates to this combination, making it a promising option for treating severe infections, like bacteremia and pneumonia, burdened with substantial unmet needs.

Recently, in vitro studies have highlighted the strong activity of SD against CRAB, noting that durlobactam significantly lowers the MICs for sulbactam and may overcome resistance mechanisms that have made other therapies ineffective. A study on Greek isolates [77] showed that SD was able to inhibit over 87.9% of CRAB isolates at MICs of ≤4/4 µg/mL. Furthermore, the combination has shown favorable results in clinical settings, significantly reducing mortality compared to other treatments, such as colistin, which is often reserved as a last-line choice owing to its nephrotoxicity and limited efficacy against resistant organisms. In vivo studies in mouse models have revealed that the proportion of the dosing interval during which the sulbactam concentration exceeds the minimum inhibitory concentration (% fT > MIC) is critical for its efficacy [78,79]. Moreover, in murine thigh and lung infection models, significant reductions in bacterial counts were observed, with durlobactam ensuring the sustained activity of sulbactam by inhibiting key β-lactamases produced by CRAB.

A key study showed that in order to achieve a 1- and 2-log reduction in the bacterial count, sulbactam had to maintain a fT > MIC above 50% of the time, which was feasible by adding durlobactam [80,81].

In May 2023, the US Food and Drug Administration (FDA) approved SD for treating CRAB hospital-acquired pneumonia (HABP) and VABP in patients 18 years and older following the ATTACK trial (phase 3, randomized trial) [82]. The latter included 181 patients with HABP, VABP, or bloodstream infections, randomly assigned to either SD or colistin. The study consisted of two parts: part A, an assessor-blind, randomized study comparing the efficacy and safety of SD with colistin, both combined with imipenem/cilastatin, and part B, an open-label observational study for individuals intolerant or resistant to colistin/polymyxin B. Results showed a 28-day all-cause mortality rate of 19% in the SD group versus 32% in the colistin group, meeting non-inferiority criteria. Nephrotoxicity was significantly lower with SD (13% vs. 38%). Serious adverse events occurred in 40% of the SD group and 49% of the colistin group, whereas treatment-related discontinuations were 11% for SD and 16% for colistin.

Safety and efficacy were also evaluated in a phase 2, double-blind, randomized, placebo-controlled study in patients with complicated urinary tract infections (cUTI) and acute pyelonephritis. Eighty patients were randomized to receive intravenous sulbactam/durlobactam 1 g/1 g or placebo with imipenem/cilastatin 500 mg every 6 h for 7 days (up to 14 days in patients with bacteremia). Overall, study success rates in the m-MITT population at the test of cure (TOC) were 76.6% and 81.0%, respectively [83].

The resistance of *A. baumannii* to SD is primarily attributed to the production of MBLs such as NDM-1, which can degrade the β-lactam ring of antibiotics, including the combination of sulbactam and durlobactam [24]. This enzymatic activity directly contributes to resistance by breaking down the antibiotic before it can inhibit cell wall synthesis in bacteria.

Additionally, mutations at specific sites within the PBP3 can reduce the binding affinity of sulbactam, thus diminishing its effectiveness at inhibiting cell wall synthesis. For instance, an A515V substitution has been associated with higher SDMICs, indicating a strong resistance mechanism [47,77]. Table 1 summarizes the information about innovative approaches.

##### Cefepime/Zidebactam

Cefepime/zidebactam (FEP-ZID) is an innovative combination antibiotic therapy showing significant promise in the treatment of infections caused by *A. baumannii*, particularly those resistant to traditional carbapenems [84]. Cefepime, a fourth-generation cephalosporin, is known for its broad-spectrum antibacterial activity, which is significantly enhanced by zidebactam, a non-beta-lactam β-lactamase inhibitor [85]. Together, they target PBPs and inhibit β-lactamases, extending the spectrum of cefepime against β-lactam-resistant bacterial pathogens comprising MDR strains of *A. baumannii* [84,86,87,88,89]. Research, including in vitro and in vivo studies, demonstrates that FEP-ZID can overcome resistance mechanisms that typically limit the efficacy of other antibiotics [90]. For instance, the [68,70,71,72,73] studies performed by Thomson et al. [91] demonstrate that cefepime combined with zidebactam holds robust activity against a panel of MDR clinical isolates, including *A. baumannii*, offering a potentially vital alternative to existing antibacterial agents. In clinical trials, FEP-ZID exhibited superior efficacy compared to some standard treatments, managing to inhibit a significant proportion of *A. baumannii* strains that were resistant to other antibiotics [92]. This includes its application in serious hospital-acquired infections, where it addresses the dire need for effective treatments against CARB strains [68,70,71,72,73].

Concerning clinical experimentation, four trials of FEP-ZID have been completed. The first, a phase 1, randomized, double-blind, placebo-controlled study concluded in 2016, has evaluated the safety, tolerability, and pharmacokinetics of escalating doses of intravenous FEP-ZID in 20 healthy adults. Participants received 3 g of zidebactam and 6 g of cefepime or placebo over 60 min every 8 h; however, results are not yet available (NCT02707107, [93]). In phase 1, the open-label study assessed single-dose pharmacokinetics in 48 subjects with varying renal function, focusing on maximum plasma concentrations and the area under the plasma concentration–time curve; also, in this case, results are still pending (NCT02942810, [94]). In 2018, a phase 1, multiple-dose, open-label study of 36 healthy adults determined and compared plasma and intrapulmonary concentrations of FEP-ZID, particularly in epithelial lining fluid and alveolar macrophages (NCT03630094, [95]). Additionally, a phase 1, randomized, double-blind, double-dummy, placebo, and positive-controlled crossover study evaluated the effect of FEP-ZID on the QT/QTc interval in healthy volunteers to estimate the incidence of delayed cardiac repolarization (NCT03554304, [96]). Currently, a phase 3, randomized, double-blind, multicenter, non-inferiority study is enrolling 528 hospitalized adults affected by cUTI or acute pyelonephritis to evaluate the efficacy, safety, and tolerability of FEP-ZID compared to meropenem with primary objectives focusing on test-of-cure success and treatment-emergent adverse events (NCT04979806, [97]).

##### Imipenem/Cilastatin/Funobactam

Imipenem/cilastatin/funobactam (formerly XNW4107) is a novel BLIC composed of imipenem, a broad-spectrum carbapenem, cilastatin, a dehydropeptidase I inhibitors, preventing the degradation of imipenem, and funobactam, a novel diazabicyclooctane BLI, which enhances this combination by neutralizing β-lactamases produced by resistant bacteria, thus restoring the efficacy of imipenem [98,99]. Although funobactam does not have direct antibacterial activity, it effectively inhibits both serine and MBL, including Ambler Class A, C, and D β-lactamases such as OXA-23 and OXA-24 found in *A. baumannii* [98,99]. Four phase I studies (NCT04801043, NCT04802863, NCT04787562, and NCT04482569) evaluated pharmacokinetics properties, safety, and tolerability of funobactam, alone and in combination with imipenem/cilastatin.

NCT05204368 is an ongoing multicenter, randomized, double-blind, double-dummy, comparative, phase 3 study evaluating the efficacy and safety of imipenem/cilastatin/funobactam in comparison with meropenem in hospitalized adults with cUTI, including acute pyelonephritis [100]. Furthermore, NCT05204563 is a multicenter, randomized, double-blind, comparative, phase 3 study assessing the efficacy and safety of intravenous imipenem/cilastatin/funobactam in comparison with imipenem/cilastatin/relebactam in adults with HABP or VABP [101].

##### Xeruborbactam

Xeruborbactam is a novel β-lactamase inhibitor that exhibits a unique mechanism of action and a broad antibiotic spectrum, particularly against Gram-negative bacteria [85]. It contains a benzoxaborole moiety, which enhances its ability to bind and inhibit a wide range of β-lactamases, including both serine- and MBL. This feature enables xeruborbactam to restore the efficacy of β-lactam antibiotics against highly resistant bacterial strains by blocking the enzymes responsible for antibiotic degradation [102].

The intrinsic antibacterial activity of xeruborbactam, although modest, is particularly notable against CRAB. This makes it a valuable candidate for addressing infections caused by MDR organisms where few treatment options exist. Sun et al. [103] highlight its efficacy when used in combination with other antibiotics, indicating that xeruborbactam can significantly extend the antimicrobial spectrum of partnered β-lactams through inhibition of critical resistance mechanisms. Moreover, Nelson et al. have demonstrated in vitro that xeruborbactam significantly enhanced the potency of meropenem against carbapenem-resistant Enterobacterales-producing serine and MBLs [104].

Given its broad-spectrum β-lactamase inhibition and the growing threat of *A. baumannii* infections, xeruborbactam is poised to play a crucial role in modern antimicrobial therapy. It has been granted Qualified Infectious Disease Product (QIDP) status, underscoring its potential importance in clinical settings facing serious bacterial resistance challenges. This status is particularly relevant for tackling infections like those caused by *A. baumannii*, which are notorious for their resilience against standard treatments.

Xeruborbactam has completed two phase 1 studies (NCT04380207 and NCT04578873) [105,106]. Being administered by the intravenous or oral route (as a prodrug form), it was found to be safe and well tolerated at doses of 1 g/day or less and resulted in exposures that exceeded non-clinical pharmacokinetic-pharmacodynamic targets [107,108,109]. Another phase 1, randomized, double-blind, single-dose, drug–drug interaction study aimed at determining the impact of co-administration of xeruborbactam on the pharmacokinetics of QPX2014 (an undisclosed β-lactam antibacterial) in healthy adult subjects was completed in 2022, but the results have not yet been published (NCT05072444, [110]).

Moreover, a phase 1, open-label, randomized, double-blind, controlled, multiple-dose pharmacokinetic and safety study of xeruborbactam oral prodrug in combination with ceftibuten in 72 healthy adults is currently ongoing. The objective is to evaluate the safety, tolerability, and pharmacokinetics of single and multiple doses of xeruborbactam oral prodrug and ceftibuten, both in combination and alone. In addition, this study will evaluate whether there are any pharmacokinetic interactions between xeruborbactam oral prodrug and ceftibuten when co-administered (NCT06079775, [111]). A second study in progress is a phase 1, open-label, single-dose study to evaluate the safety and pharmacokinetics of ceftibuten/xeruborbactam oral prodrug in 32 participants with renal impairment (NCT06157242, [112]).

#### 3.2.2. Phage Therapy

Phage therapy, which utilizes bacteriophages to target and lyse specific bacterial pathogens, is gaining traction as a viable treatment option, especially pertinent for combating CRAB [113]. Phages approach was explored by Hua et al. [114] using neutropenic mouse model to evaluate the efficacy in treating CRAB-induced lung infections. The study utilized a newly isolated lytic phage, SH-Ab15519, which significantly improved survival rates in mice, demonstrating the potential of phages as effective agents against CRAB infections [114].

Recent studies have highlighted the efficacy of various phage therapies against MDR *A. baumannii*, demonstrating significant advancements in both clinical and preclinical settings. For instance, phage YMC 13/03/R2096 ABABBP and molar φ-R2096 have shown high lytic activities in dose-dependent manners against *A. baumannii* growth. Specifically, phage SH-AB15519, isolated from hospital wastewater, has proven to be effective in treating pneumonia caused by CRAB in mice, presenting a potential alternative to traditional antibiotics due to its lack of virulence or antimicrobial resistance genes [115]. Additionally, phage therapies like φ KM18P have not only improved survival rates in *A. baumannii* bacteremia models in mice but also reduced inflammatory responses, further validating the potential of phage therapy as a viable treatment option [115]. These findings underscore the potential of phage therapy, in combination with antibiotics, to serve as a crucial tool in combating bacterial infections, particularly those resistant to conventional treatments.

#### 3.2.3. Other Strategies

*LPS Inhibitors*: a novel class of macrocyclic peptides that inhibits LPS transport with a potent activity against MDR *Acinetobacter* strains in mouse infection models have recently been reported and under investigation [116,117,118,119]. Zosurabalpin is an experimental macrocyclic antibiotic which primary mechanism of action for involves the inhibition of the LPS transport system within CRAB [119].

Zosurabalpin specifically targets a protein complex known as LptB2FGC, which is integral to the LPS transport process [119]. This complex acts as a molecular machine that shuttles LPS molecules across the periplasmic space to the outer membrane. By binding to this complex, zosurabalpin disrupts the normal function of the LptB2FGC complex, causing LPS molecules to accumulate within the bacterial cell. This accumulation of LPS within the cell is lethal to the bacteria, as it effectively disrupts the integrity of the bacterial cell membrane, leading to cell death. Research has shown that zosurabalpin’s ability to interfere with the LPS transport is highly specific for *A. baumannii*, which means it should not affect other types of bacteria. This specificity is advantageous because it reduces the likelihood of the drug disrupting the microbiota [118]. Furthermore, the unique target of zosurabalpin reduces the potential for cross-resistance with other antibiotics, as resistance mechanisms developed against other antibiotics are unlikely to affect zosurabalpin [116,117,118,119].

Zosurabalpin has demonstrated impressive efficacy in preclinical studies, effectively killing CRAB strains in laboratory settings and in infected animal models. These studies have shown that zosurabalpin can prevent CRAB-induced sepsis in mice, suggesting its potential effectiveness in treating severe human infections caused by this superbug [116,117,118,119]. Currently, zosurabalpin is undergoing phase 1 clinical trials to evaluate its safety and efficacy in human patients [116,117,118,119].

*Protein synthesis inhibitors*: apramycin (EBL-1003) with one phase 1 study [NCT05590728], an aminoglycoside-derivate antibiotic, and zifanocycline (KBP-7072) with four phase 1 studies [NCT02454361, NCT05507463, NCT04532957, NCT02654626], a third-generation tetracycline, showed promising efficacy against CRAB, both targeting 30S ribosomal subunit, owing the ability to both evade modifying enzymes and to bind to a modified drug target site [120,121,122]. Becker et al. demonstrated apramycin efficacy against *A. baumannii* lung infections in mice models, and pharmacokinetic studies showed an efficient lung penetration ratio of 88% [122].

Li et al. showed potent in vitro activity of zifanocycline against *A. baumannii* isolates, with MIC values ranging from 0.06 to 0.5 mg/L, demonstrating significant bactericidal efficacy in neutropenic murine thigh infection model [123].

*Novel polymyxins*: three molecules, MRX-8, QPX9003, and SPR206, are being assessed with phase 1 studies [NCT04649541, NCT04808414, and NCT04868292, respectively]. All of them showed the same mechanism of action, consisting in binding LPS and disrupts the Gram-negative cell envelope, and reduced toxicities compared to “old” polymyxin (such as colistin). All of them showed in vitro activities against CRAB strains [124,125,126].

*Bacterial cell division inhibitors*: filamentous temperature-sensitive protein Z (FtsZ), a prokaryotic cytoskeleton protein, plays a crucial role in bacterial cell division by forming a dynamic Z-ring at the center of the cell, which initiates the division process [127]. As an essential component for bacterial cytokinesis, any disruption in FtsZ function can result in the failure of cell division and subsequent bacterial death [128]. This protein is highly conserved across eubacteria, archaea, and chloroplasts, making it an attractive broad-spectrum target for antibacterial agents. Despite sharing some structural and functional similarities with eukaryotic tubulin, significant differences allow for the development of FtsZ-specific inhibitors that minimally affect human cells. Targeting FtsZ presents a novel antibacterial mechanism crucial in the face of increasing antibiotic resistance. Researchers have identified numerous natural and synthetic FtsZ inhibitors, such as cinnamaldehyde and PC190723, which demonstrate significant antibacterial activity and potential for therapeutic development [129].

## 4. Conclusions

The relentless rise of *A. baumannii* as a formidable nosocomial pathogen poses a significant challenge to contemporary healthcare systems worldwide. Its capacity to develop and disseminate resistance to multiple antimicrobial agents, especially carbapenems, underlines the urgent need for innovative therapeutic approaches and robust infection control strategies. The detailed exploration of its resistance mechanisms in this paper—from enzymatic degradation and target modification to reduced permeability and biofilm formation—provides critical insights into the molecular and cellular intricacies that confer this pathogen’s high resilience against standard treatments.

Given the complexity and dynamism of *A. baumannii* resistance mechanisms, our collective response must be equally sophisticated and adaptive. This includes the continued development and clinical implementation of new antimicrobial agents such as cefiderocol, and innovative combinations like SD and FEP-ZID, which show promise in overcoming pathogen resistance strategies. Furthermore, the potential of phage therapy offers a complementary approach that could be particularly useful in cases where traditional antibiotics fail. Deep-learning approaches and artificial intelligence-based methods could play a significant role against antimicrobial resistance by discovering antibiotic alternatives, offering a fast, cost-efficient, and labor-independent strategy that reduces the chances of failure at later stages of drug discovery [130,131,132].

In light of the findings presented it seems obvious that combating *A. baumannii* requires a multifaceted strategy that integrates advanced scientific research, clinical innovation, and rigorous public health measures. This entails not only developing new drugs but also enhancing diagnostic techniques to rapidly identify resistant strains, optimizing antimicrobial stewardship programs to prevent the emergence of resistance, and implementing stringent infection control protocols to curb the spread of this pathogen in healthcare settings.

The battle against *A. baumannii* is not confined to the microcosm of microbiological research or hospital wards; it is a broader public health dilemma that demands a coordinated, global response. By fostering collaboration among researchers, clinicians, and policymakers, and by investing in research and development, aiming to find new antibiotics and alternative strategies, we can fortify our defenses against *A. baumannii* and ensure that effective treatment options remain available for those in need.

Eventually, the fight against *A. baumannii* is not just a battle to be fought but a war to be avoided. Our true strength lies not in combating these resilient foes head-on but in outmaneuvering them with diligent prevention, such as infection control measures and antibiotic stewardship programs.

## Figures and Tables

**Figure 1 ijms-25-06814-f001:**
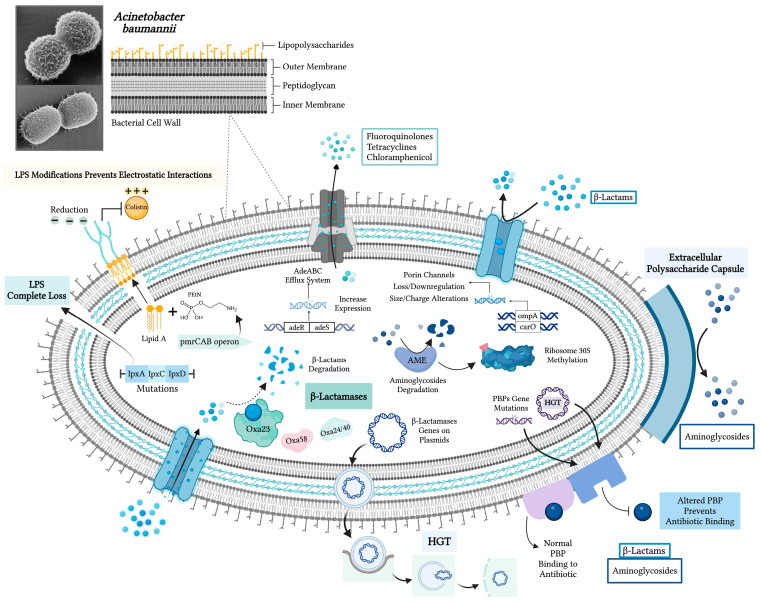
*Acinetobacter baumannii* resistance mechanisms. Abbreviations: AME—aminoglycoside-modifying enzyme; carO—carbapenem resistance-associated outer membrane protein; HGT—horizontal gene transfer; LPS—lipopolysaccharide; Oxa23, 58, 24/40—OXA-type carbapenemases; ompA—outer membrane protein A gene; PBP—penicillin-binding protein; PEtN—phosphoethanolamine (created with BioRender.com, accessed on 18 June 2024).

**Figure 2 ijms-25-06814-f002:**
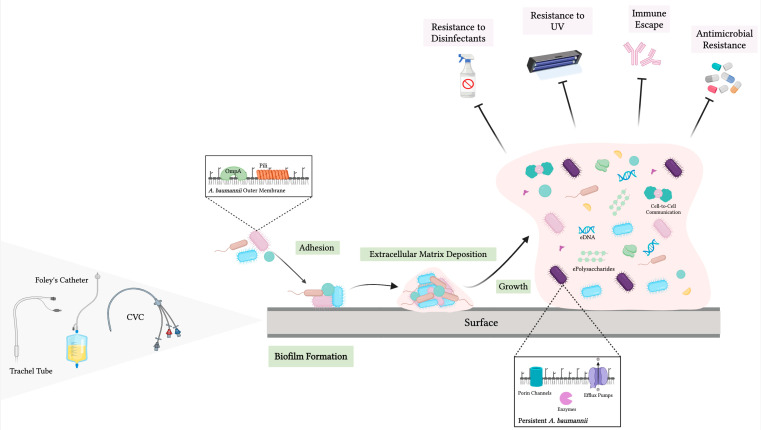
*Acinetobacter baumannii* biofilm characteristics and associated resistance (created with BioRender.com, accessed on 20 May 2024).

**Table 1 ijms-25-06814-t001:** Emerging pharmacotherapies and related clinical trials against MDR *Acinetobacter baumannii*.

Name of the Drug	Mechanism of Action	Clinical Trials
Sulbactam/durlobactam (SD, Xacduro^®^)	β-lactamase inhibitor combined with a non-beta-lactam beta-lactamase inhibitor.	1. A randomized, active-controlled study to evaluate the efficacy and safety of intravenous sulbactam/etx2514 in the treatment of patients with infections caused by *Acinetobacter baumannii-calcoaceticus* complex (ATTACK) (NCT03894046). 2. A double-blind, randomized, placebo-controlled study to evaluate the safety and efficacy of intravenous sulbactam/etx2514 in the treatment of hospitalized adults with complicated urinary tract infections, including acute pyelonephritis (NCT03445195).
Cefepime/zidebactam (FEP-ZID; WCK5222)	Fourth-generation cephalosporin that disrupts the synthesis of the peptidoglycan layer combined with a non-β-lactam β-lactamase inhibitor.	1. A randomized, double-blind, placebo-controlled study to evaluate the safety, tolerability, and pharmacokinetics of multiple escalating doses of intravenous WCK 5222 (zidebactam and cefepime) in healthy adult human subjects (NCT02707107). 2. A phase 1, open-label, single-dose study to investigate the pharmacokinetics of intravenous WCK 5222 (FEP-ZID) in patients with renal impairment (NCT02942810). 3. A phase 1, multiple-dose, open-label study to determine and compare plasma and intrapulmonary concentrations of WCK 5222 (cefepime and zidebactam) in healthy adult human subjects (NCT03630094). 4. A randomized, double-blind, double-dummy, placebo- and positive-controlled, crossover study to evaluate the effect of WCK 5222 on the QT/QTc interval in healthy volunteers (NCT03554304). 5. A phase 3, randomized, double-blind, multicenter, comparative study to determine the efficacy and safety of cefepime/zidebactam vs. meropenem in the treatment of complicated urinary tract infection or acute pyelonephritis in adults (NCT04979806).
Imipenem/cilastatin/funobactam	Broad-spectrum carbapenem, combined with a dehydropeptidase I inhibitor and a novel diazabicyclooctane.	1. Four phase 1 studies evaluated pk properties, safety, and tolerability of funobactam, alone and in combination with imipenem/cilastatin (NCT04801043, NCT04802863, NCT04787562, and NCT04482569). 2. An ongoing multicenter, randomized, double-blind, double-dummy, comparative, phase 3 study evaluating the efficacy and safety of imipenem/cilastatin/funobactam vs. meropenem in hospitalized adults with complicated urinary tract infections, including acute pyelonephritis (NCT05204368). 3. A multicenter, randomized, double-blind, comparative, phase 3 study assessing the efficacy and safety of iv imipenem/cilastatin/funobactam vs. imipenem/cilastatin/relebactam in adults with hospital-acquired bacterial pneumonia or ventilator-associated bacterial pneumonia (NCT05204563).
Xeruborbactam (XER, QPX7728)	Ultra-broad-spectrum cyclic boronate inhibitor of serine and MBL.	1. A phase 1, randomized, double-blind, placebo-controlled, ascending single and multiple-dose study of the safety, tolerability, and pharmacokinetics of intravenous (IV) QPX7728 alone and in combination with QPX2014 in healthy adult subjects (NCT04380207). 2. A phase 1, randomized, double-blind, placebo-controlled, ascending single- and multiple-dose study of the safety, tolerability, and pharmacokinetics of oral QPX7831 in healthy adult subjects (NCT04578873). 3. A phase 1, randomized, double-blind, single-dose, drug–drug interaction study to determine the impact of co-administration of QPX7728 on the pharmacokinetics of QPX2014 in healthy adult subjects (NCT05072444). 4. A phase 1, open-label, drug–drug interaction, and randomized, double-blind, controlled, multiple-dose pharmacokinetics and safety study of xeruborbactam oral prodrug (QPX7831) in combination with ceftibuten in healthy adult participants (NCT06079775). 5. A phase 1, open-label, single-dose study to determine the safety and pharmacokinetics of oravance (ceftibuten/xeruborbactam oral prodrug [QPX7831]) in participants with renal impairment (NCT06157242).
Zosurabalpin (RG6006)	Macrocyclic peptides that inhibit LPS transportation.	1. Phase 1, multicenter, single-dose, uncontrolled, open-label (NCT05614895). 2. Phase 1, randomized, sponsor-open, adaptive, single- and multiple-ascending dose, placebo-controlled study (NCT04605718).
Apramycin (EBL-1003)	Amynoglicoside derivate.	A phase 1, open-label study to evaluate the plasma PK profile of apramycin and lung penetration of apramycin in epithelial lining fluid and alveolar macrophages after a single IV apramycin dose in healthy subjects, a secondary objective is to assess safety and tolerability (NCT05590728).
Zifanocycline (KBP-7072)	Third-generation tetracycline.	Four phase 1 studies assessing safety and tolerability (NCT02454361, NCT05507463, NCT04532957, NCT02654626).
MRX-8 QPX9003 SPR206	Polymixins.	Phase 1 studies to assess safety and tolerability (NCT04649541, NCT04808414, and NCT04868292).

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
