# Peer review of "Unveiling the Secrets of Acinetobacter baumannii: Resistance, Current Treatments, and Future Innovations"

_ijms, 2024, doi:10.3390/ijms25136814_

Round 1

Reviewer 1 Report

Comments and Suggestions for Authors

Dear reviewers, I had the pleasure of reading your manuscript entitled "Unveiling the Secrets of Acinetobacter baumannii: Resistance, Current Treatments, and Future Innovations".  I believe that the manuscript is of interest to the clinical and scientific community and has relevant and current information; however, I have some comments.

-        Line 30: Please check whether all bacterial names are in italics.

-        Lines 72 and 77 check the style of references in all texts according to the author’s instructions of IJMS.

-        Line 137 I think this subtitle "2.1.2. Aminoglycoside-modifying enzymes does not correspond to the 2.1. Enzymatic Degradation.

-        Line 232 I think this subtitle " Active Efflux" does not correspond to type 2.3. Reduced permeability.

-        Line 310, 317 and others please add a space between the number and units.

-        Line 320-321 should be “extensive drug-resistant strains” and insert the abbreviation here XDR and erase in line 328.

-        Table 1 has not been cited in the manuscript.

-        Please check all references, someone is incomplete (reference 9), and bacteria names should be in italics.

I hope that these comments will be helpful in improving the manuscript.

Author Response

Line 30: Please check whether all bacterial names are in italics.

Reply: We checked what you suggested.

Lines 72 and 77 check the style of references in all texts according to the author’s instructions of IJMS.

Reply: We changed what you suggested, in accord to journal guidelines.

Line 137 I think this subtitle "2.1.2. Aminoglycoside-modifying enzymes does not correspond to the 2.1. Enzymatic Degradation.

Reply: We changed the title of section 2.1 to “Enzymatic inactivation”, which better fits with all the subtitles

 Line 232 I think this subtitle " Active Efflux" does not correspond to type 2.3. Reduced permeability.

Reply: We changed the title of section 2.3 to “Reduced permeability and Active efflux”, which better fits with all the subtitles

 Line 310, 317 and others please add a space between the number and units.

Reply: Done

Line 320-321 should be “extensive drug-resistant strains” and insert the abbreviation here XDR and erase in line 328.

Reply: We changed what you suggested

Table 1 has not been cited in the manuscript.

Reply: We cited the table 1 within the text

Please check all references, someone is incomplete (reference 9), and bacteria names should be in italics.

Reply: We changed what you pointed out

Reviewer 2 Report

Comments and Suggestions for Authors

This paper reviewed the resistance mechanisms of Acinetobacter baumannii and explored the current and novel therapeutic options against carbapenem-resistant A. baumannii (CRAB). The combination sulbactam-durlobactam, cefepime-zidebactam, xeruborbactam, and the newest zosurabalpin, represented the most promising alternatives under investigation. Moreover, the phage therapy offers a complementary potential approach. The topic fits the scope of this journal, and may benefit the design of more effective agents for the prevention and treatment of these diseases caused by this resilient pathogens. The manuscript is well-organized, and the references are mostly updated. The key issues are required to be addressed before its publication on Int. J. Mol. Sci.

Major points:

1. The novel therapeutic approach targeting the cell division protein FtsZ of A. baumannii are suggested to be involved, and the following references are suggested to be cited (Li, et al. Advances in the discovery of novel antimicrobials targeting the assembly of bacterial cell division protein FtsZ. Eur J Med Chem, 2015, 95, 1-15. Chai, et al. Cinnamaldehyde derivatives act as antimicrobial agents against Acinetobacter baumannii through the inhibition of cell division. Frontiers in Microbiology, 2022, 13, 967949.).

2. In the section of treatment options, the antibacterial activities and the available pharmaceutical properties of these drugs/drug candidates are suggested to be included, especially for the readership in basic research in drug discovery and development.

Minor points:

1. In Figure 1, the smallest font is suggested to be enlarged for the better readability.

2. The chemical structures of the representative drugs/drug candidates are suggested to be included in this manuscript for the better readability and larger readership.

3. In the conclusion section, the artificial intelligence (AI) technology in the discovery of antibacterial is suggested to be involved as one of the perspectives (Talat, et al. Artificial intelligence as a smart approach to develop antimicrobial drug molecules: A paradigm to combat drug-resistant infections. Drug Discovery Today, 2023, 28(4), 103491.).

Comments on the Quality of English Language

The English language is ok.

Author Response

Major points:

  1. The novel therapeutic approach targeting the cell division protein FtsZ of A. baumannii are suggested to be involved, and the following references are suggested to be cited (Li, et al. Advances in the discovery of novel antimicrobials targeting the assembly of bacterial cell division protein FtsZ. Eur J Med Chem, 2015, 95, 1-15. Chai, et al. Cinnamaldehyde derivatives act as antimicrobial agents against Acinetobacter baumannii through the inhibition of cell division. Frontiers in Microbiology, 2022, 13, 967949.).

Reply: We added the reference you suggested. Furthermore, we have improved our discussion about new molecules against CRAB to enhance the completeness of the review.

  1. In the section of treatment options, the antibacterial activities and the available pharmaceutical properties of these drugs/drug candidates are suggested to be included, especially for the readership in basic research in drug discovery and development.

Reply: Thank you for your valuable opinion. We have improved both the section on current options and the section on future and innovative strategies, hoping that this makes the paper more complete, useful, and attractive for readers.

Minor points:

  1. In Figure 1, the smallest font is suggested to be enlarged for the better readability.

Reply: We changed the figure as you suggested. In addition, we changed the page layout in horizontal to better visualize the image.

  1. The chemical structures of the representative drugs/drug candidates are suggested to be included in this manuscript for the better readability and larger readership.

Reply: Thank you for your suggestion. We considered your point, but we believe that including too many chemical structures would make the paper difficult to read and cause confusion. Therefore, we decided not to add the chemical structures and hope you understand our decision. Additionally, this is beyond the scope of this review.

  1. In the conclusion section, the artificial intelligence (AI) technology in the discovery of antibacterial is suggested to be involved as one of the perspectives (Talat, et al. Artificial intelligence as a smart approach to develop antimicrobial drug molecules: A paradigm to combat drug-resistant infections. Drug Discovery Today, 2023, 28(4), 103491.).

Reply: We added a few lines about AI in the conclusion section citing the paper you suggested.